# Analogues of Pyrimidine Nucleosides as Mycobacteria Growth Inhibitors

**DOI:** 10.3390/microorganisms10071299

**Published:** 2022-06-27

**Authors:** Liudmila A. Alexandrova, Anastasia L. Khandazhinskaya, Elena S. Matyugina, Dmitriy A. Makarov, Sergey N. Kochetkov

**Affiliations:** Engelhardt Institute of Molecular Biology RAS, 32 Vavilov Str., 119991 Moscow, Russia; khandazhinskaya@bk.ru (A.L.K.); matyugina@gmail.com (E.S.M.); dmitmakarov_97@mail.ru (D.A.M.); snk1952@gmail.com (S.N.K.)

**Keywords:** mycobacteria, *Mycobacterium tuberculosis*, drug resistance, nucleoside analogues, inhibitors, thymidine monophosphate kinase, thymidylate synthase

## Abstract

Tuberculosis (TB) is the oldest human infection disease. Mortality from TB significantly decreased in the 20th century, because of vaccination and the widespread use of antibiotics. However, about a third of the world’s population is currently infected with *Mycobacterium tuberculosis* (*Mtb*) and the death rate from TB is about 1.4–2 million people per year. In the second half of the 20th century, new extensively multidrug-resistant strains of *Mtb* were identified, which are steadily increasing among TB patients. Therefore, there is an urgent need to develop new anti-TB drugs, which remains one of the priorities of pharmacology and medicinal chemistry. The antimycobacterial activity of nucleoside derivatives and analogues was revealed not so long ago, and a lot of studies on their antibacterial properties have been published. Despite the fact that there are no clinically used drugs based on nucleoside analogues, some progress has been made in this area. This review summarizes current research in the field of the design and study of inhibitors of mycobacteria, primarily *Mtb*.

## 1. Introduction

Tuberculosis (TB) is the oldest human infectious disease, which has probably been inherited from early hominids [1]. Currently, about a third of the world’s population is infected with *Mycobacterium tuberculosis* (*Mtb*). The death rate from TB is 1.4–2 million people a year. Tuberculosis affects more than nine million people every year, and since 2000 the incidence rate has fallen by only 1.5 percent a year [2,3].

The development of drugs for the treatment of TB has remained one of the priority tasks of pharmacology and medicinal chemistry since the identification of *Mtb* by R. Koch in 1882 [4]. During the 20th century, many anti-tuberculosis drugs were discovered or synthesized and numerous treatment regimens were developed using first-line and second-line antibiotics [5,6,7]. As a result, the mortality from this disease was reduced tenfold [2,3] and it seemed that the final victory over TB had been achieved.

However, in the middle of the last century, drug-resistant forms of TB were identified. A rapid spread of antibiotic-resistant strains of *Mtb* was observed, on which standard chemotherapy has practically no effect [8,9,10,11,12,13,14,15,16,17]. In the future, worsening of the situation should be expected. According to WHO, by 2050, out of 10 million deaths that could be associated with drug resistance, about a quarter will be due to drug-resistant strains of tuberculosis [2,3]. There is an opinion that humanity is entering a post-antibiotic era, in which even common infections or minor injuries can be life-threatening [18,19].

Thus, the emergence of new multidrug-resistant strains is one of the main problems in the current century [2,8]. There is an acute need for the development of novel drugs that can act on new pathogen’s targets and are active against resistant strains of bacteria and viruses.

Nucleosides as antibacterial agents, in particular anti-tuberculosis, look very attractive. Nucleosides and nucleotides are the basic structural units of DNA and RNA which are involved in the biosynthesis of proteins. They also act as cofactors in different metabolic pathways, including lipid and polyamine biosynthesis [19,20]. It is known that in some cases even small modifications of the heterocyclic base or carbohydrate fragment of the nucleoside can have a significant effect on the recognition and inhibition of certain targets and thus block the growth of the corresponding microorganisms.

The antibacterial activity of nucleoside derivatives was revealed not so long ago; systematic studies of antimycobacterial activity have been carried out over the past two decades. Several reviews have been published on modified nucleosides that inhibit the growth of mycobacteria (*Mtb*, *Mycobacterium bovis* and *Mycobacterium avium*) [19,21,22,23,24,25,26,27]. Modified nucleosides inhibit enzymes involved in the metabolism of purines [26,27,28] and pyrimidines [23,24,25,26,27,29] and in the processes of iron transportation necessary for the life of mycobacteria [21,22,30] and others. Despite the fact that there is no data on clinical trials of the nucleoside agents in the literature, further study of their antimicrobial properties is forthcoming and may reveal new antibiotics of a nucleoside nature.

The present review describes the antimycobacterial activity of modified pyrimidine nucleosides and their analogues as well as possible targets of their action. It considers only derivatives and analogues of pyrimidine nucleosides, since there are detailed reviews about the anti-tuberculous activity of purine nucleoside derivatives, mainly about the synthesis and study of sulfamoyl-adenosine analogues, inhibitors of the biosynthesis of siderophores—iron carriers—which are necessary for the survival of mycobacteria [21,22,30,31].

## 2. Inhibitors of *M. tuberculosis* Growth with Unidentified Target

Kumar et al. firstly synthesized nucleoside analogues and accessed their ability to inhibit the growth of mycobacteria in vitro [32]. They obtained derivatives of ribo-, deoxyribo- and dideoxyribonucleosides in order to study the effect of various groups at C-5, C-2 and C-3 positions of the nucleoside on the antimycobacterial activity against *M. avium*. The authors suggested that such compounds are able to interact with mycobacterial enzymes involved in the process of nucleic acid synthesis as inhibitors or competitive substrates, thus affecting the synthesis of DNA or RNA [32,33], but this hypothesis was not confirmed. Among the wide range of synthesized compounds, 5-substituted derivatives **1**–**4** showed the highest activity (Figure 1). Compounds selectively inhibited the growth of *M. avium* by 90% in 1–5 µg/mL concentrations, which is the same level of values for clinically used rifampicin (MIC_90_ 2 µg/mL) [32]. However, analogue **4** was completely inactive against *M. bovis* and *M. tuberculosis* [33].

Since 2005, the same group has published a series of papers [34,35,36,37]. They have synthesized a wide range of C-5 alkynyl pyrimidine nucleoside derivatives that effectively inhibit the growth of mycobacteria, including *M. tuberculosis* (Figure 2) [35,36,37]. The synthesis of these compounds was carried out by the Sonogashira reaction starting from the corresponding 5-iodo derivatives of 2′-deoxyuridine, modified at the carbohydrate moiety [38,39].

The SAR studies showed that 5-(1-dodecynyl) and 5-(1-tetradecinyl) derivatives of deoxynucleosides had the best antimycobacterial activity. 5-Substituted derivatives of 2′-deoxyribouridine were more effective than their riboanalogues, while 5-modified uracil did not inhibit the growth of mycobacteria [34]. The study of SAR of modification of the carbohydrate fragment as a possible antimycobacterial pharmacophore of these nucleosides showed that almost all 2′-deoxy-**5** [34], 2′,3′-dideoxy-**6,** 2′,3′-dideoxy-3′-fluoro-**7 [36]**, 2′,3′-dideoxy-2′-fluoro-uridines **9**, arabinouridines **8** [35] and acyclic nucleosides **11** [37] with long 1-alkynyl substituents are able to inhibit the growth of mycobacteria.

Thus, among the alkynyl derivatives of 2′-deoxynucleosides **5a**–**h** and **10a**–**e**, the highest activity against *M. bovis* and *M. avium* was shown by 5-dodecynyl-**5f** (MIC_90_ 50 μg/mL), 5-tetradecinyl-2′-deoxyuridine **5g** (MIC_90_ 10 µg/mL) and 5-dodecynyl-2′-deoxycytidine **10d** (MIC_90_ 10 µg/mL). It should be noted that the activity increases with the length of the substituent in the 5 position of the pyrimidine base. To elucidate the role of the carbohydrate fragment, the authors also obtained 2′,3′-dideoxy derivatives **6a**–**f** and 3′-fluoro-2′,3′-dideoxy derivatives **7a**–**f**. The best activity against *M. bovis*, *Mtb* and *M. avium* was shown by compounds **6c** (MIC_50_ 1–2, 1–2, 1–50 µg/mL for each mycobacterium, respectively) and **7e** (MIC_50_ 1, 1–2, 100 μg/mL, respectively). In most cases, the studied compounds were non-toxic for different cell cultures (Huh-7, Vero or HFF cell cultures: CC_50_ ≥ 100 µg/mL). Thus, in general, with an increase in the length of the hydrocarbon radical, it leads to an increase in activity including one against the rifampicin-resistant *Mtb* strain, which is also observed for nucleosides with different modifications of the carbohydrate moiety. Among derivatives **8a**–**e** and **9a**–e, compounds **10c** (MIC_90_ 1–5, 1–5, 50–100 µg/mL) and **9d** (MIC_90_ 1, 1, 50 µg/mL) showed the highest activity against *M. bovis*, *Mtb* and *M. avium* (MIC_90_ 1, 1, 50–100 μg/mL) [34,35,36,37], which confirms the importance of the long alkyl substituent.

Later various 2′- or 3′-halogen derivatives of pyrimidine nucleosides containing substituents at the C-5 position of the base were synthesized by Canadian scientists [40]. Among this series, the most effective inhibitor was a 3′-bromo derivative **12** (Figure 3) with a MIC_50_ 1–3 µg/mL against the strains of *Mtb* H37Rv and H37Ra. Interestingly, compound **12** also inhibits the intracellular form of *Mtb*on, a human monocyte cell line (THP-1) infected with H37Ra at a concentration of 25 μg/mL, showing a higher activity against the intramacrophage form of mycobacterium compared to the extra macrophage form. In addition, no cytotoxicity was found for this type of compound against human cells (CD_50_ > 100–200 µg/mL) [40].

Further studies led to the synthesis and study of the antimycobacterial properties of a series of 3′-azido-, 3′-amino- and 3′-halogen derivatives of pyrimidine nucleosides, among which only 3′-azido-5-ethyl-2′,3′-dideoxyuridine **13** showed significant activity against *M. bovis*, *M. avium* and *Mtb* (MIC_50_ 1–5 µg/mL) (Figure 3) [41]. In the same year, Kumar et al. published paper on the synthesis and study of 5-ethyl-, 5-hydroxymethyl- and 5-methoxymethyl-substituted pyrimidines. The best inhibitory activity in this series of nucleosides was shown by 2′-fluoro derivative **14** (MIC_50_ 5 µg/mL), as well as 4-thio derivative **15** (Figure 3), inhibiting the *Mtb* H37Rv strain (MIC_50_ 0.5 µg/mL), H37Ra (MIC_50_ 1 µg/mL), as well as *M. bovis* and *M. avium* (MIC_50_ 1 and 10 µg/mL, respectively) [41]. No cellular cytotoxicity was found for these compounds up to the highest concentration tested (CC_50_ > 100 µg/mL).

Recently, series of 5-alkynyl-substituted uridines **16** [42] and 6-methyluridines **17** [43] were synthesized (Figure 4) via Sonogashira reaction. It was found that a methyl group in C-6 position of the pyrimidine ring had no impact on the yields of target compounds as well as on the inhibitory activity. Synthesized nucleosides showed activity against *M. bovis* and *Mtb* at concentrations of 1–100 µg/mL (MABA test). The MIC_50_ values of some compounds were similar or close to that of the reference drug rifampicin.

Of considerable interest are works on the synthesis of 5-alkyloxymethyl and 5-alkyltriazolylmethyl derivatives of pyrimidine nucleosides with long alkyl substituents (Figure 5) [44,45]. The anti-tuberculosis activity of the synthesized compounds was tested on two *Mtb* strains: laboratory strain H37Rv and multidrug-resistant (MDR) strain MS-115 (resistant to five first-line anti-TB drugs). C-5 Alkynyl derivatives **18c**, **23b**, and **23c** with a hydrocarbon chain length of 10 or 12 carbon atoms were the most effective; while compounds with shorter substituent lengths weakly inhibited the growth of mycobacteria. In general, 2′-deoxyuridine derivatives were more active than 2′-deoxycytidine derivatives. Analogues **18c**, **23b** and **23c** inhibited the growth of the *Mtb* at concentrations comparable to currently used drugs: laboratory strain H37Rv at MIC_99_ 20, 10, 10 µg/mL, respectively and resistant MS-115 at MIC_99_ 50, 10, 10 µg/mL, respectively [44,45]. The value of CD_50_ in A549, *Vero* and *Jurkat* cell cultures was about 70–100 μg/mL.

Described compounds containing long hydrophobic groups are poorly soluble in water-organic solutions. At the same time, better soluble 5-alkyltriazolylmethyl-2′-deoxyuridines with short alkyl residues, in contrast to decyl- and dodecyl-triazolylmethyl-2′-deoxyuridines, demonstrated antibacterial activity against a number of Gram-positive bacteria [46]. On the other hand, 2′-deoxyuridine derivatives containing long hydrophilic substituents **26a**,**b** in the 5-position (which is similar to a chain of 10–12 atoms in the C-5 position of the pyrimidine base of the nucleosides that previously showed the best anti-tuberculosis activity) were well-soluble compounds, but completely lost antimicrobial activity [47].

A number of 5-modified 6-aza-2′-deoxyuridines **27a**,**b** and **28a**,**b** (Figure 6) demonstrated inhibitory activity not only against *Mycobacterium smegmatis* (MIC_99_ 0.2–0.8 mM), but also against *Staphylococcus aureus* (MIC_99_ 0.03–0.9 mM). Compound **28b** showed the best results, comparable with the activity of clinically used drugs [48].

### 5′-Norcarbocyclic Uracil Derivatives

Another interesting example of antimycobacterial nucleoside analogues are 5-alkynyl 5′-norcarbocyclic uridine derivatives [49], which were synthesized by the method described earlier [34]. Carbocyclic nucleosides are analogues of natural nucleosides in which the oxygen atom in the furanose ring is replaced with methylene group. A number of carbocyclic nucleosides, such as aristeromycine and neplanocin A, have been found in nature [50,51,52,53]. Carbocyclic analogues are more stable to phosphorylases and hydrolases but due to their structural similarity to natural nucleosides are recognized by many receptors and enzymes. Carbocyclic nucleosides showed a wide spectrum of biological activity, in particular, antiviral and anticancer properties [50,51,52,53,54]. However, in some cases being converted to triphosphate forms, which are very similar to natural nucleoside triphosphates, carbocyclic analogues exhibit significant toxicity as a result of their undesirable recognition by ATP-metabolizing enzymes [55,56,57]. In 5′-norcarbocyclic nucleoside analogues, the primary hydroxyl group is replaced by a secondary one, which prevents their recognition and phosphorylation by cellular kinases and leads to much less toxicity [53,57,58,59,60]. In this regard, 5′-norcarbocyclic nucleosides are a promising class of compounds for drug development.

The synthesized 5′-norcarbocyclic uracil derivatives (Figure 7) demonstrated anti-tuberculosis activity against both the laboratory (H37Rv) and multidrug-resistant (MS-115) *Mtb* strains. All compounds, except for 1-(4-hydroxy-2-cyclopenten-1-yl)-5-tetradecynyluracil (**29c**), showed moderate inhibitory activity on the laboratory strain H37Rv (MIC_99_ 20–40 µg/mL). The compound **29c** as racemic mixture completely suppressed the growth of mycobacteria at MIC_99_ 10 µg/mL. Individual enantiomers of the compound **27c** were isolated using enzymatic separation with Amano PS lipase [61] and tested against MDR strain (MS-115). The racemic mixture **29c** as well as (+)**29c** enantiomer inhibited the growth of the resistant strain MS-115 at MIC_99_ 10 μg/mL. At the same time, it was shown that the (−)**29c** enantiomer inhibited the growth of laboratory strain H37Rv at MIC_99_ 5 μg/mL [49]. Compound **29c** had no toxicity in *Vero* cell cultures up to the highest tested concentration (CC_50_ > 125 µg/mL).

Later, 5-alkyloxymethyl and 5-alkyltriazolylmethyl 5′-norcarbocyclic uridine analogues **30** and **31** were synthesized (Figure 8) [62]. The antibacterial activity of these compounds was assessed against a number of bacteria, including both Gram-positive and Gram-negative strains. Such uracil derivatives prove to be selective inhibitors towards mycobacteria, demonstrating significant activity in vitro against two *M. smegmatis* strains (MIC_99_ 67 and 6.7–67 μg/mL for mc2155 VKPM Ac 1339 strains), attenuated *Mtb*ATCC 25177 (MIC_99_ 28–61 µg/mL) and *M. bovis* ATCC 35737 (MIC_99_ 50–60 µg/mL) and two strains of *Mtb*, including the laboratory strain H37Rv (MIC_99_ 20–50 µg/mL) and MDR MS-115 strain resistant to five first-line anti-TB drugs (MIC_99_ 20–50 µg/mL). However, unlike both 5-alkyloxymethyl and 5-alkyltriazolylmethyl derivatives of 2′-deoxyuridine [44,45] and 5-alkynyl 5′-norcarbocyclic uracil derivatives [49], analogues **30** and **31** turned out to be toxic to human monocyte cells [62].

Another group of 5′-norcarbocyclic uridine analogues as *Mtb* inhibitors (MIC_99_ 5–40 μg/mL) based on 5-arylaminouracil derivatives was synthesized [63,64]. The compounds showed no cytotoxicity in Vero, A549 and *Huh7* cell cultures up to a concentration of 50 μg/mL.

Thus, the best activities among uridine analogues (MIC_90_ 1–10 mg/mL) were found for 5-alkynyl (decynyl, dodecynyl, tetradecynyl, pyridylethynyl, etc.) uracil derivatives bearing various modifications in the carbohydrate moiety [23,32,33,34,35,36,37,44,45,47]. At the same time, 2′-deoxypyrimidine derivatives with long 5-alkyloxymethyl or 5-alkyltriazolylmethyl substituents (5-dodecyloxymethyl-2′-deoxyuridine, 5-decyltriazolylmethyl-2′-deoxyuridine, 5-dodecyltriazolylmethyl-2′-deoxycytidine, etc.) were able to inhibit in vitro the growth of both the laboratory strain H37Rv and the multidrug-resistant clinical isolate *Mtb*MS-115 at MIC_99_ about 10–20 mg/mL [44,45,47]. Two groups of carbocyclic nucleoside analogues, namely, 5-alkynyl- and 5-alkoxymethyl-5′-norcarbocyclic uridine analogues also showed high activity against *Mtb* strains H37Rv and MDR MS-115 [49,62]. The observed biological activity of these new 5-substituted uracil derivatives well correlates with previously published data on the inhibition of *Mtb* growth by nucleoside analogues containing 5-substituted pyrimidine bases and various sugars, as well as their corresponding acyclic or carbocyclic analogues [23,32,33,34,35,36,37,47,49,62]. It can be assumed that these compounds have the same mechanism of antimicrobial action, and their effectiveness depends primarily on the length and structure of the substituent at the C-5 position of the pyrimidine base.

## 3. N4-Modified Cytidine Derivatives

Development of the effective inhibitors of mycobacteria among pyrimidine nucleosides have suggested that N4-modified cytidine derivatives containing extended lipophilic alkyl substituents can also exhibit antibacterial activity [65,66]. As shown earlier [23,32,33,34,35,36,37,47,49,62], derivatives with extended lipophilic alkyl substituents had the best activity against mycobacteria among the C-5-substituted pyrimidine nucleosides. To test the hypothesis, linear alkyl substituents (-C_n_H_2n+1_, *n* = 8–14) were introduced into the N4 position of the cytosine residue (Figure 9). A number of synthesized derivatives demonstrated inhibitory activity against *M. smegmatis* and some Gram-positive bacteria, including drug-resistant strains of *Staphylococcus aureus* MRSA [65]. It was shown that 2′-deoxy-5-methylcytidine derivatives **33** containing extended N-alkyl substituents C_10_H_21_ (**33a**) and C_12_H_25_ (**33b**) have the best activity (24–50 μM) comparable in some cases with the action of clinically used antibiotics.

N4-alkyl-2′-deoxy-5-methylcytidines (**34**) with additional modifications of the carbohydrate fragment also demonstrated high antibacterial activity (27–110 μM). However, these compounds, containing amino and alkylamino groups in the 3′-position of the carbohydrate fragment, showed high cytotoxicity on the HeLa cell line (6–60 μM), and couldn’t be considered as potential antibacterial drugs [66].

## 4. Depot Forms

### 4.1. Depot Forms with Increased Solubility

Low water solubility is a serious problem during the study of the antibacterial and/or antiviral properties of modified nucleosides. One of the ways to overcome this obstacle is the synthesis of depot forms (“prodrugs”) (The term depot form means a biologically inert or weakly active compound containing the parent drug, which undergoes transformation in vivo due to chemical or enzymatic cleavage to release the parent drug), which can be used to improve the pharmacokinetics, pharmacodynamics, solubility and toxicity of the drug. This approach helps in the effective delivery of the drug to the target of action and improves its pharmacokinetic characteristics due to the slower release of the active component [67,68,69,70,71,72,73].

In order to develop a new set of antibacterial nucleosides, a representative library [74,75,76] of 3′- and 5′-tri- or tetraethylene glycol prodrugs of 5-alkyloxymethyl- and 5-alkyltriazolylmethyl-2′-deoxyuridines with described anti-tuberculosis activity were synthesized (Figure 10) [44,45].

The compounds were two orders of magnitude more soluble compared to the parent nucleosides, had significant inhibitory activity against a number of bacteria, including drug-resistant strains of *M. smegmatis* (as well as *S. aureus*), and exhibited low cytotoxicity [74,75,76]. The replacement of the triethylene glycol residue in C-5′ position by tetraethylene glycol led to an increase in water solubility and a significant decrease in inhibitory activity. It can be assumed that the inhibitory activity of 5-alkyloxymethyl-2′-deoxyuridines (**35** and **36**) depends on the substituent in C-5 position of the heterocyclic base and changes in the row: C_10_H_21_ < C_11_H_23_ < C_12_H_25_ > C_14_H_29_ > C_16_H_33_. The data obtained and the convenience of the proposed method for the synthesis of glycol prodrugs of nucleoside derivatives indicate the potential of their use as antibacterial agents.

### 4.2. Dual Action Nucleoside Derivatives

TB is the main opportunistic infection in HIV-infected people, which greatly raises the risk of HIV infection progression and death. HIV infection increases the chance of activation of the latent form of *Mtb* [77], and at the same time causes the rapid development of the disease soon after (re)infection [78]. *Mtb* and HIV act synergistically, which leads to a decrease the immunological status and death without treatment [79]. Therefore, it seems attractive to search for new dual-acting drugs that simultaneously inhibit the development of TB and HIV.

In this regard, five new compounds that can simultaneously inhibit HIV and tuberculosis in vitro and ex vivo have been designed and synthesized.

These compounds are heterodimers of 5-arylaminouracil derivative [63] or analogue of 2′-deoxyuridine **23b** as anti-TB molecules [45] and of 3′-azido-3′-deoxythymidine (AZT) as anti-HIV molecules. The compounds described herein are the first examples of nucleoside derivatives having both anti-HIV and anti-TB activity.

Heterodimers **39**–**43** (Figure 11) are depot forms of two active agents either with fast (**39**–**41**) or slow (**42**–**43**) hydrolysis rates. To optimize the structure of the active components and the linker that binds them, further studies are needed. The evaluation of the pharmacokinetic parameters of anti-HIV and anti-TB drugs released into the blood of laboratory animals should be conducted. It is worth noting that compound **39** showed the highest activity and inhibited HIV and the drug-resistant strain of tuberculosis MS-115 better than the original nucleoside **23b** and a mixture of AZT and **23b** (1:2 mol-equiv.). In MT-4 cell culture, moderate cytotoxicity and cytostatic effect were observed for heterodimers **39**, **40**, **42** and **43**. However, the most active compound (**39**) had no toxicity and low cytostatic effect (about 4 times lower than AZT itself).

Thus, heterodimer **39** could be the candidate for future optimization and studying in vivo.

## 5. Target Enzymes for Nucleoside Inhibitors of *M. tuberculosis*

For some of the pyrimidine nucleoside derivatives that are inhibitors of mycobacteria growth, cellular targets were identified [29,80]. These are thymidylate synthase (TS) and thymidine monophosphate kinase (TMPK), catalyzing the biosynthesis of thymidine 5′-mono- and diphosphates (Figure 12).

Thymidine monophosphate kinase (TMPK) belongs to the nucleoside monophosphate kinase superfamily and catalyzes the phosphorylation of thymidine monophosphate (TMP) to thymidine diphosphate (TDP) using ATP as a donor of phosphate group. This enzyme connects the de novo and recycling pathways for the synthesis of thymidine triphosphate (Figure 12). By the de novo pathway, TMP is synthesized from deoxyuridine monophosphate (dUMP) by thymidine synthase (TS) (ThyA or ThyX); by the recycling pathway, thymidine is phosphorylated by thymidine kinase to form TMP [29,81].

To date, a significant number of thymidine derivatives have been synthesized and tested as inhibitors of *Mtb* growth. For some of them, the action on the enzymes mentioned above is directly shown, for others the target was not finally identified.

### 5.1. Thymidine Monophosphate Kinase Inhibitors

TMPK is a perspective target for the development of anti-tuberculosis drugs that are able to selectively effect the mycobacterial isoform of TMPK, which is vital for the mycobacteria growth.

TMPK^mtb^ is a homodimer (Figure 13), each monomer of which consists of 214 amino acid residues. TMPK^mtb^ is a thermostable enzyme (Tm 68°) (which is also characteristic of other *Mtb* proteins). The spatial structure of TMPK^mtb^ is similar to other enzymes of this class, but significantly differs in amino acid sequences (26% homology with *E. coli* TMPK, 25% with yeast TMPK and 22% with human TMPK) [82,83].

### 5.2. Nucleic Base Modification

The replacement of the methyl group at the 5 position with a bromine atom resulted in 5-Br-dUMP (**44**), which is an active substrate of TMPK^mtb^ (**44**, *Km* = 30 µM, Table 1) [83,84]. The introduction of bulky substituents into the 5 position led to the loss of substrate properties and resulted in inhibitory activity of these compounds (the most active were 5-hydroxymethyl-(**45**) and 5-furan-2-yl dUMP (**46**) (*Ki* 110 and 140 μM respectively) [85] (Table 1). Modification at the 2 position of the pyrimidine ring led to 5-methyl-iso-dCMP, which turned out to be a competitive inhibitor of the enzyme with a *Ki* of 130 μM [83].

### 5.3. Modification of the Carbohydrate Fragment

Immediately after obtaining the crystal structure of TMPK^mtb^ [83], it was found that 3′-azido-2′,3′-dideoxythymidine 5′-monophosphate (**47**) is an inhibitor of this enzyme (*Ki* 10 µM) [83,84]. It is probable that TMPK^mtb^ does not phosphorylate **47** due to the interaction of the terminal nitrogen atom of the azido group with the active site of the enzyme, which may lead to weaker binding to ATP [85,86,87].

As a rule, structural modifications of **47** led to a decrease in inhibitory activity, for example, 3′-aminothymidine monophosphate **48** showed only insignificant inhibitory activity, while the 3′-fluoro analogue was a substrate with *Km* of 30 μM [86].

Later, a number of 3′-nucleotides with branched chains were synthesized: 3′-C-azidomethyl-**49**, 3′-aminomethyl-**50**, 3′-fluoromethyl-**51** and 3′-hydroxymethyl-**52** derivatives of dTMP (*Ki* 10.5—29 μM) [88], which proves the ability of the enzyme to bind inhibitors with bulky substituents in the 3′ position. The absence of a phosphate group in these derivatives increased their selectivity, but led to loss of activity [88,89].

**Table 1 microorganisms-10-01299-t001:** Interaction of thymidine derivatives with TMPK^mtb^.

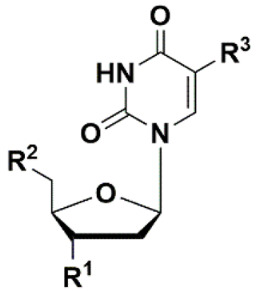					
Compound	R^1^	R^2^	R^3^	Ki (µM) TMPK^mtb^	Ref.
**dTMP**	OH	OPO_3_^2−^	CH_3_	4.5	[83,84]
**Thymidine**	OH	OH	CH_3_	27	[83,84]
**44**	OH	OPO_3_^2−^	Br	30 *	[83,84]
**45**	OH	OPO_3_^2−^	CH_2_OH	110	[85]
**46**	OH	OPO_3_^2−^	Furan-2-yl	140	[85]
**47**	N_3_	OPO_3_^2−^	CH_3_	10	[83,84,85,86]
**48**	NH_2_	OPO_3_^2−^	CH_3_	235	[86]
**49**	CH_2_N_3_	OPO_3_^2−^	CH_3_	12	[88]
**50**	CH_2_NH_2_	OPO_3_^2−^	CH_3_	10.5	[88]
**51**	CH_2_F	OPO_3_^2−^	CH_3_	15	[88]
**52**	CH_2_OH	OPO_3_^2−^	CH_3_	29	[88]
**53**	OH	N_3_	CH_3_	7	[89]
**54**	OH	NH_2_	CH_3_	11	[89]
**55**	OH	OH	Br	5	[90]
**56**	OH	OH	Cl	10	[90]
**57**	N_3_	OH	Br	10.5	[91]
**58**	N_3_	OH	Cl	16	[91]

* Substrate, *Km* (μM).

Some derivatives without a 5′-monophosphate group proved to be effective inhibitors of TMPK^mtb^: 5′-azido-**53** and 5′-aminothymidine **54** (which showed the highest activity) [90], 5-bromo-**55** and 5-chloro-2′-deoxyuridine **56** and their 3′-azido analogues **57** and **58** [91] (Table 1). *Ki* values for these compounds were close to the *Km* of the natural substrate. So, azido- and amino-modifications seem attractive for the design of potent selective enzyme inhibitors.

Representatives of another class of nucleoside analogues demonstrating effective and selective inhibition of TMPK^mtb^ were bicyclic thymidine derivatives **59**–**61** (Figure 14). The synthesis of these compounds and their analogues, carried out by Van Calenbergh and co-workers [89,92], was discussed in detail in the earlier published reviews [24,25].

The urethane and thiourethane derivatives **59a**,**b** showed *Ki* values of 13.5 and 3.5 μM on isolated TMPK^mtb^. However, they were inactive against *M. tuberculosis*, probably due to the low permeability of the bacterial cell wall for polar molecules. To solve this problem, a 5′-deoxyanalogue of the thiourethane derivative **60** was synthesized, which proved to be the most effective inhibitor of TMPK^mtb^ (*Ki* 2.3 μM), showing, however, low activity against *Mtb* (MIC_99_ 31 µg/mL, 100 μM). Compound **61**, which is a five-membered analogue of **59b**, was noticeably less active against both the enzyme (*Ki* 8 µM) and mycobacteria (MIC_99_ 500 µg/mL, 1.6 mM) [89].

Revealed inhibitory activity of dinucleoside **62** was unexpected, since the binding site for the second nucleoside is absent in TMPK^mtb^. Further studies, however, showed that such a compound can act as a competitive inhibitor of the two substrates required for the enzyme: TMP and ATP. This promising activity of analogue **62** has pushed further synthesis of derivatives capable of increasing the inhibitory properties.

During the optimizing of the structure, one of the two monomers was replaced by different aromatic groups in order to enhance the electron-withdrawing properties of the molecule [93]. As a result, a series of 3′-arylthiourea thymidine derivatives **63a**–**g** was synthesized (Figure 14). According to computer modeling data [89], the authors assumed that the 5′-derivatives of α-thymidine will bind to TMPK^mtb^ similar to their β-analogues with substituents at the 3′-position, because of similarity in spatial structure. In order to confirm this hypothesis, a number of α-thymidine derivatives **64a**–**d** were synthesized (Figure 15).

To synthesize both series of nucleosides, thymidine anomers were converted into the corresponding 3′- or 5′-amino derivatives, followed by the addition of the necessary aryl isothiocyanates. The synthesis of these compounds is described in detail in reviews [24,25].

The study of the inhibitory properties of 3′-modified β-thymidine derivatives showed that the substituents in the phenyl ring significantly affect the activity of the compounds, leading to the highest efficiency in the case of electron-withdrawing groups. The best inhibitor in this series was the 3-trifluoromethyl-4-chlorophenyl derivative **63g** (*Ki* 5.0 μM). 5′-Modified α-thymidine derivatives were more active than their β-analogues. The same time lipophilic and electron-withdrawing substituents in the phenyl fragment significantly increased the inhibitory activity of the compounds. Compound **64c** was the most active against TMPK^mtb^ with a *Ki* value of 0.6 μM, a selectivity index of 600 and effective inhibition of *M. bovis* growth (MIC_99_ 20 μg/mL, 50 μM). Analogue **64c** was able to inhibit the *Mtb* growth by 39% maximum at 6.25 µg/mL concentration [93].

A number of arylpyrimidines have been investigated as inhibitors of TMPK^mtb^ [94,95,96,97]. Initial screening results revealed derivative **66**, which showed the best inhibition of TMPK^mtb^ (*Ki* 42 μM) (Figure 16).

Further modifications of this compound, primarily the linker between the nucleic base and the naphthalimide fragment, led to a significant increase in activity (*Ki* 0.42 and 0.27 μM for compounds **67** and **68**, respectively). (Figure 16). However, none of the synthesized compounds inhibited the growth of *M. bovis* BCG and *Mtb* cells at concentrations up to 7.5 and 32 μg/mL (19 and 80 μM), respectively.

Thus, to date, a wide range of TMPK^mtb^ inhibitors have been obtained, some of which have demonstrated a significant inhibitory effect on *Mtb*. So, this enzyme can serve as a convenient model in the search for new inhibitors of mycobacteria growth.

### 5.4. Thymidylate Synthase Inhibitors

The thymidylate synthase enzyme (ThyA) is essential for living organisms to synthesize thymidine monophosphate de novo [98]. ThyA catalyzes the reductive methylation of 2′-deoxyuridine 5′-monophosphate (dUMP) to thymidine 5′-monophosphate (TMP).

Further studies have shown that some microorganisms do not have ThyA encoding genes, but remain viable in an environment with a lack of thymidine. Such microorganisms are capable of expressing the ThyX enzyme. This enzyme and the encoding gene *ThyX* are rare in eukaryotes and are completely absent in humans [99,100,101].

ThyX and ThyA catalyze the same reaction; however, these enzymes do not have structural similarity and have different mechanisms of catalysis. Despite ThyA, ThyX activity depends on the coenzyme flavin adenine dinucleotide (FADH_2_) as a hydride ion donor (Figure 17) [102,103,104,105].

*Mtb* strains are known to contain both ThyA and ThyX enzymes and the latter is absolutely necessary for mycobacteria to survive in macrophages. Since macrophages are the reservoir of *Mtb* in the latent phase of the disease, ThyX inhibitors can become prototypes of drugs that act on this phase. The suggestion was made based on the analysis of the crystal structure of the ThyX enzyme (Figure 18) [102,104,106].

5-Fluoro-2′-deoxyuridine 5′-monophosphate **69** (Figure 19) was the first compound, which inhibits the ThyX enzyme with an IC value of 0.57 µM. However, this compound also inhibited ThyA at the same range (IC_50_ 0.29 µM) [101].

A series of papers on selective inhibitors of ThyX was published by Piet Herdewijn and co-workers [105,107,108]. Among the obtained nucleotides containing alkynyl or aryl substituents at the C-5 position of the heterocyclic base, 5-(3-octanoylaminoprop-1-ynyl)-2′-deoxyuridine 5′-monophosphate **70** (Figure 19) had the highest activity. This compound inhibited ThyX (IC_50_ 0.91 μM) more effectively than ThyA (IC50 > 50 μM). Despite the noticeable activity and selectivity of the inhibitor, it was impossible to study its antibacterial properties due to the high polarity of the compound caused by the presence of the phosphate group, which limits the penetration through the cell wall of mycobacteria. In order to increase pharmacological properties, McGuigan and co-workers designed and synthesized so-called “ProTide” prodrug forms of the target compound [109]. This strategy consists in “masking” the negative charges of the monophosphate group with two lipophilic residues: an amino acid ester and an aryloxy fragment [110]. Synthesis of such prodrugs increases the possibility of penetration into the cell by passive diffusion and stability to dephosphorylation. The release of the original monophosphate occurs in two stages: enzymatic and subsequent spontaneous chemical hydrolysis. This strategy has been successfully applied in the development of anticancer drugs [111,112] and antiviral nucleoside analogues [113].

ProTide derivatives **73** (Figure 19) proved to be the most effective inhibitors of *Mtb* growth (strain H37Rv), showing moderate antimycobacterial activity (MIC_99_ from 62.5 to 250 mg/L, 83–324 μM) [109].

Herdewijn et al. synthesized a series of potential inhibitors of thymidylate synthase and thymidine monophosphate kinase, namely 5′-monophosphates of 5-modified 6-aza-2′-deoxyuridines **71** (Figure 19) [107]. However, the compounds were inactive against ThyX; only derivative **71b** showed weak inhibitory activity (40.9% inhibition of ThyX and 13.8% inhibition of ThyA, respectively, at a concentration of 50 μM) [107].

Since the 5′-phosphates of nucleotides are easily cleaved by esterases, similarly to the successful development of antiviral nucleoside analogues [114,115], a series of acyclic nucleoside phosphonates (ANP) containing 5-alkynyluracil (**70**, Figure 19) and a number of others were synthesized [108]. An important advantage of ANPs is their catabolic stability. From all tested compounds, only (6-(5-(3-octanamidoprop-1-yn-1-yl)-2,4-dioxo-3,4-dihydropyrimidin-1(2H)-yl)hexyl)phosphonate **72** showed 43% inhibition of ThyX at 50 µM. So, the data indicate some possibility of ThyX inhibition by representatives of this class of acyclic nucleoside phosphonates [108].

Agrofoglio and co-workers [116] have developed a new class of ANPs based on the mimic of two natural cofactors (dUMP and FAD) as inhibitors against *M*tb ThyX (Figure 20).

Twelve analogues of acyclic nucleoside phosphonates were synthesized and tested for their activity against *Mtb* ThyX. These phosphonates were 5- and 7-substituted quinazoline and 5-substituted uracil linked at N1 to a (E)-but-2-enyl phosphonic acid moiety. Only the quinazoline analogue **75b** showed a poor 31.8% inhibitory effect on ThyX at 200 μM. Nevertheless, based on **75b**, future development is needed to develop an optimal acyclic chain and identify more potent compounds.

To check the hypothesis about a potential target of anti-TB action, a group of 5′-monophosphates of 2′-deoxyuridine derivatives was synthesized and their inhibitory activity against ThyX was evaluated. Extended alkyl substituents in the C-5 position were introduced through the oxymethyl (**79a**–**c**) or (1,2,3-triazol-1-yl)methyl (**79d**,**e**) linkers [117] (Figure 21). The parent nucleosides (**18** and **23**) previously demonstrated significant inhibition of *Mtb* growth [44,45].

Synthesized derivatives of 2′-deoxyuridine 5′-monophosphates **79a**–**c** and **79d,e** were tested against ThyA and ThyX according to a previously described method [105]. All compounds showed no inhibitory activity against ThyA up to concentrations of 100 μM. Inhibition of ThyX was weak and only 5-undecyloxymethyl-2′-deoxyuridine 5′-monophosphate ammonium salt **79b** showed significant inhibition of ThyX (IC_50_ 8.32 μM + 1.39 μM).

Binding of compound **79b** to the ThyX active site was modeled by docking it into the structure of *Mtb* ThyX (PDB file 2AF6) [104], ThyA (PDB file 4FOA) [104] and human TS (PDB file 3HB8) [118] using the Molecular Operating Environment (MOE) program [119]. The results of molecular modeling provided qualitatively consistent data in the systems of parameters MMFF94x [120,121] and AMBER 99 [122]. As shown on Figure 22, the substituent at the C-5 position is located within the ThyX protein core. Similar calculations showed that the compound **72b** is not able to bind to the active site of human TS, ThyA and, thus, the selectivity of the inhibitory effect of 5-alkyloxymethyl-2′-deoxyuridine. Although **79b** inhibits ThyX worse than the previously described inhibitor N-(3-(5-(2′-deoxyuridine-5′-monophosphate))prop-2-ynyl) octanamide **70** (IC_50_ 0.91 μM) [103], high yield synthesis of 5-undecyloxymethyl-dUMP (**79b**) makes it promising for further studies. Thus, to date, some selective effective inhibitors of ThyX have been synthesized.

## 6. Potential Targets of the Antimicrobial Action for 5-Modified 2′-Deoxynucleoside Derivatives and Their 5′-Norcarbocyclic Analogues

Despite the intensive research, the biological targets and mechanism of action of 5-modified 2′-deoxynucleoside derivatives and their 5′-norcarbocyclic analogues have not been finally identified. As mentioned above, Herdewijn et al. have shown that a number of 5′-monophosphates of 5-modified 2′-deoxyuridines effectively inhibit the flavin-dependent ThyX thymidylate synthase of *Mtb* (a unique enzyme for mycobacteria [99,100,104], practically not interacting with the main bacterial enzyme ThyA (also close to eukaryotic thymidylate synthases) [105,107,108,117]. So, it can be assumed that ThyX could be at least one of the possible targets of action for 5-modified 2′-deoxyuridines [105,107,108,117]. On the other hand, as mentioned above, compounds that couldn’t be phosphorylated in vitro, namely 5′-iodo-, azido- and amino-derivatives of 5-dodecyloxymethyl-2′-deoxyuridine [45] and 5-substituted carbocyclic 5′-norcarbocyclic uridine analogues with extended 1-alkynyl [49], alkyloxymethyl and alkyltriazolylmethyl [60] substituents have significant inhibitory activity against *Mtb*.

In an attempt to better understand the mode of action of 5-modified pyrimidine nucleosides, an electron microscopic study (TEM transmission electron microscopy) of mycobacteria *Mtb* (strain H37Rv), incubated with active representatives of 5-alkyloxymethyl and alkyltriazolylmethyl derivatives of 2′-deoxyuridine and its 5′-norcarbocyclic analogues, was carried out [62,123] (Figure 23).

Authors studied changes in the morphology of *Mtb* cells on (i) ultrathin sections and (ii) fixed whole bacteria stained with uranyl acetate [123]. In the control experiment, the absolute majority (~97%) of cells had a standard shape (length 2.3–4.5 μm and diameter 270–550 nm) and were surrounded by a three-layer bacterial wall ~14 nm thick. The addition of 5-substituted pyrimidine nucleosides to the incubation medium led to dramatic changes in the morphology of bacterial cells. All cells observed in randomly selected areas of ultrathin sections were divided into several groups according to their morphology, with cells with intact morphology being the minority (less than 8.5–22.3%). The number of vacuolated cells increased sharply to 31.1–37.9% in the presence of compounds **18c**, **23c**, **29c** and **80**.

Compounds **18c**, **23c**, **29c** and **80** effectively inhibited the growth of *Mtb* H37Rv cells and caused a number of morphological changes (Figure 24). Firstly, lipid accumulation: intracellular vacuole-like inclusions were observed in cells. Several electron microscopy studies have shown previously, that the accumulation of such lipid inclusions in mycobacteria allows them to survive under unfavorable conditions and during the dormant period [124]. In particular, *Mtb* cells have been shown to accumulate lipid inclusions under oxidative stress, iron deficiency, isoniazid exposure and a number of other factors, including poor treatment outcomes [125,126,127]. These data correlate well with the observations that in the control experiment, when mycobacteria cells were grown in a medium containing only DMSO/Tween-80 solvent, the proportion of vacuolated cells increased to ~19%, but there was no inhibition of cell growth. On the contrary, after treatment with studied compounds, the proportion of vacuolated cells with accumulation of lipid inclusions increased to 30–35%, and the cell wall in 41–61% of cells was destroyed or damaged.

The thick three-layered cell wall is a unique structural feature of mycobacteria. It was observed microscopically in several laboratory strains of mycobacteria and clinical isolates of *Mtb* [128,129]. The cell wall is essential for the survival of *Mtb*, as it acts as a permeability barrier to drug entry and modulates the host’s immune response [130]. Treatment of H37Rv cells with compounds **18c**, **23c**, **29c** and **80** resulted not only in protrusions and/or indentations on the surface, but also in the destruction of bacterial cells. It should be noted that the total number of lesions in *Mtb* cells after their treatment with compounds **18c**, **23c**, **29c** and **80** correlates with their anti-tuberculosis activity (MIC) values. Thus, the total number of damages caused by the action of studied compounds is 41.5; 44.7; 61.8 and 45.2 while MIC_99_ is 20; 20; 10 and 20, respectively.

As reported earlier [131,132], the anti-TB drug isoniazid causes the destruction of the H37Rv cell wall by inhibiting the synthesis of mycolic acids. However, the morphology of H37Rv cells treated with isoniazid is significantly different from H37Rv cells treated with 5-substituted pyrimidine nucleosides. It suggests different mechanisms of action of isoniazid and 5-substituted pyrimidine nucleosides.

Thus, several mechanisms of action can be suggested for 5-modified pyrimidine nucleosides against *Mtb*, including (1) inhibition of *Mtb* flavin-dependent ThyX and (2) destruction of the cell wall or inhibition of enzymes associated with its biosynthesis.

## 7. Conclusions

As mentioned above, the antimycobacterial activity of nucleoside derivatives and analogues has been revealed recently, but a number of studies on their antibacterial properties have been published. Despite the fact that there are no drugs based on the described compounds, certain successes have been achieved in this area:Two specific enzymes of *Mtb* were identified as targets for compounds with antimycobacterial activity: thymidine monophosphate kinase and thymidylate synthase Thy-X. A number of effective inhibitors were synthesized.Several series of nucleoside derivatives and analogues have been synthesized with high antimycobacterial activity, comparable to the activity of some anti-TB drugs currently used in clinic and effective against drug-resistant strains of *Mtb*. Some compounds can be considered as drug prototypes.Rational methods have been developed to increase the bioavailability of nucleoside derivatives and analogues with high antimycobacterial activity by design of their depot forms.An original approach was proposed for the synthesis of depot forms with dual-action, which was used for obtaining compounds simultaneously effective against HIV and *Mtb*.

Thus, the use of nucleoside derivatives and analogues as inhibitors of mycobacteria is of considerable interest for microbiology, biochemistry, medicinal chemistry and pharmacology.

## Figures and Tables

**Figure 1 microorganisms-10-01299-f001:**
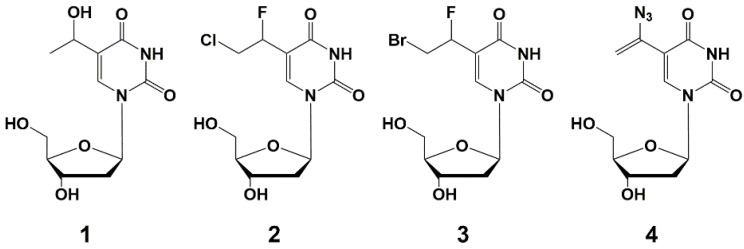
The first selective inhibitors of *M. avium* **1**–**4** among nucleoside derivatives.

**Figure 2 microorganisms-10-01299-f002:**
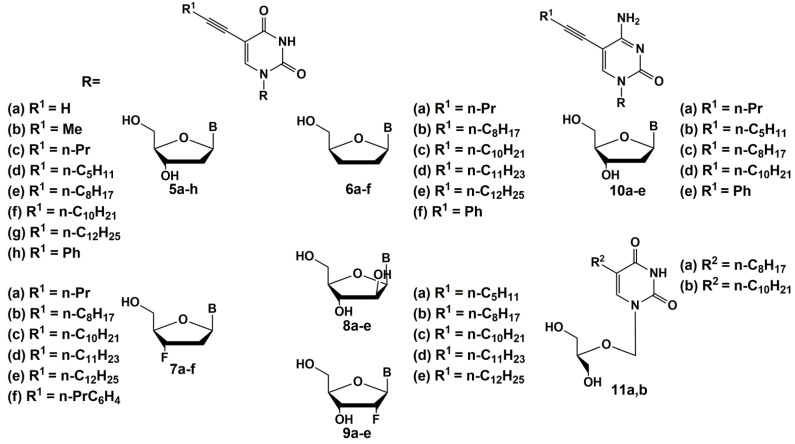
C-5 alkynyl derivatives of pyrimidine nucleosides.

**Figure 3 microorganisms-10-01299-f003:**
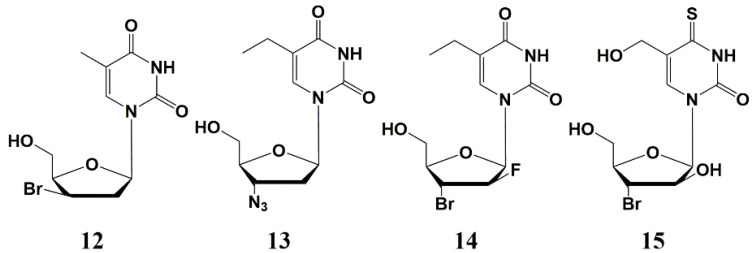
Pyrimidine nucleosides modified at the carbohydrate moiety **12**–**15**—inhibitors of mycobacteria growth.

**Figure 4 microorganisms-10-01299-f004:**
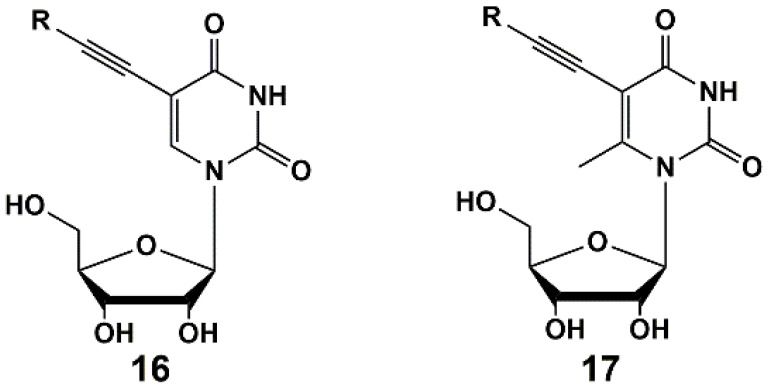
C-5 alkynyl derivatives of uridine and 6-methyluridine.

**Figure 5 microorganisms-10-01299-f005:**
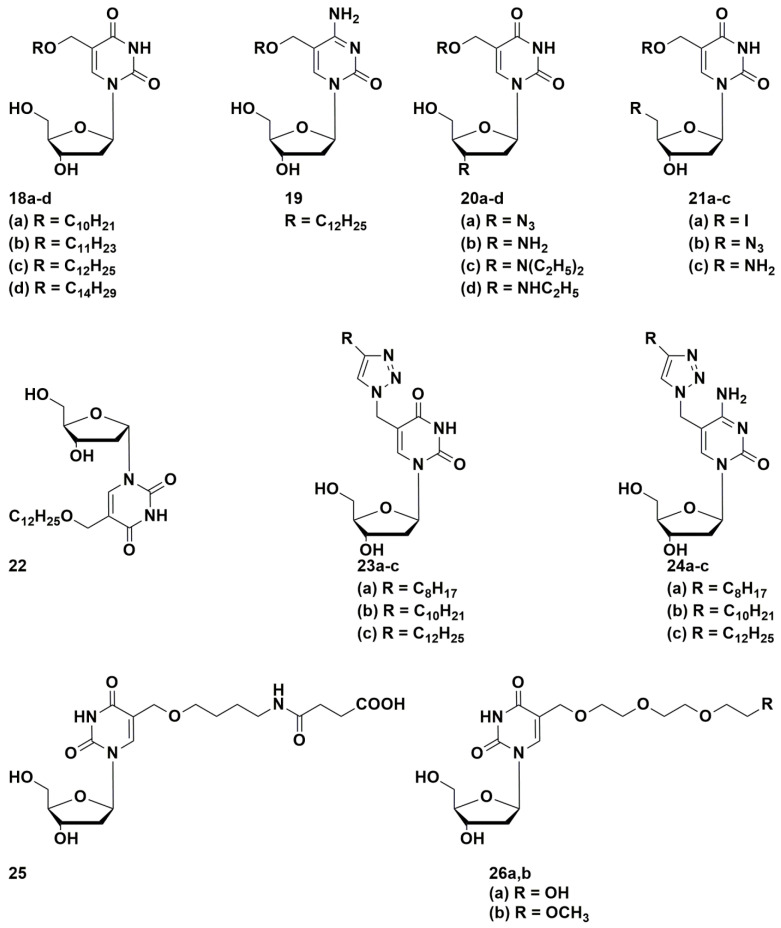
Pyrimidine derivatives of 2′-deoxy-, 3′-azido and 3′-amino-2′,3′-dideoxynucleosides [42,43].

**Figure 6 microorganisms-10-01299-f006:**
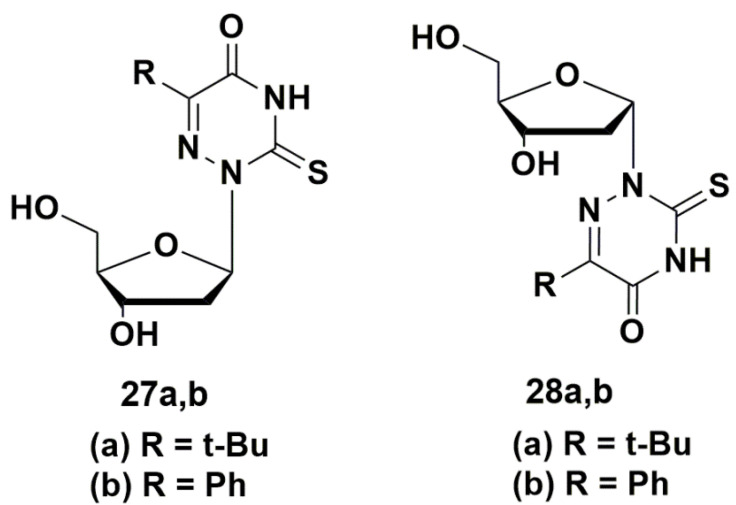
Structures of 5-modified 6-aza-2′-deoxyuridines.

**Figure 7 microorganisms-10-01299-f007:**
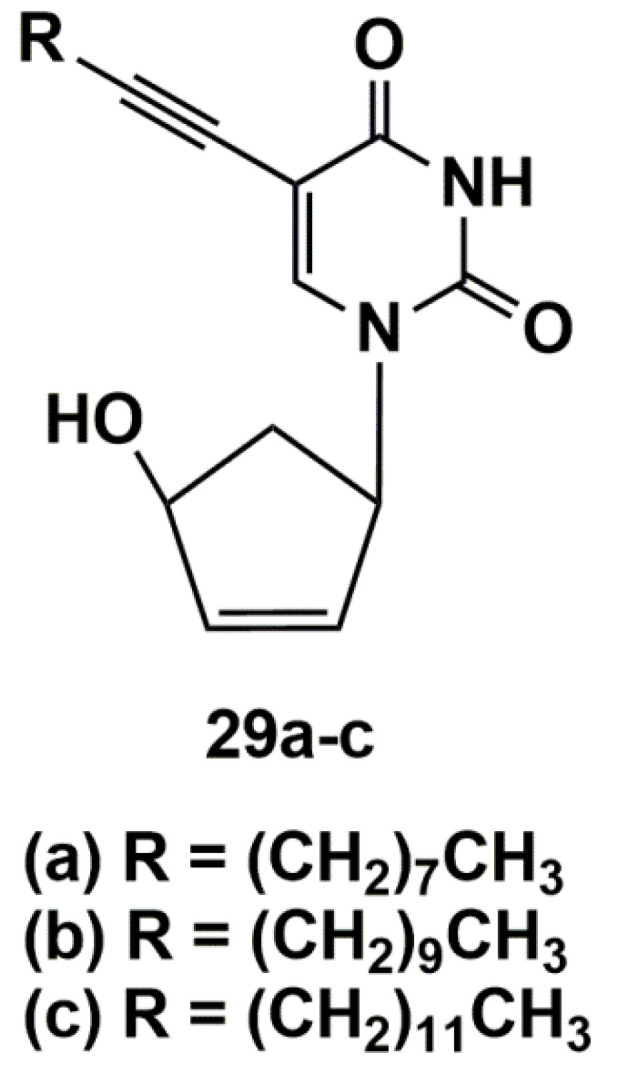
5′-Norcarbocyclic uracil derivatives.

**Figure 8 microorganisms-10-01299-f008:**
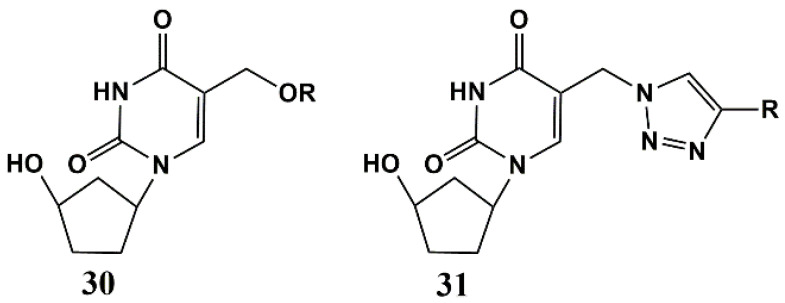
5-Alkyloxymethyl and 5-alkyltriazolylmethyl substituted 5′-norcarbocyclic uridine analogues. R: C_10_H_21_ (**30a**, **31a**), C_11_H_23_ (**30b**, **31b**), C_12_H_25_ (**30c**, **31c**).

**Figure 9 microorganisms-10-01299-f009:**
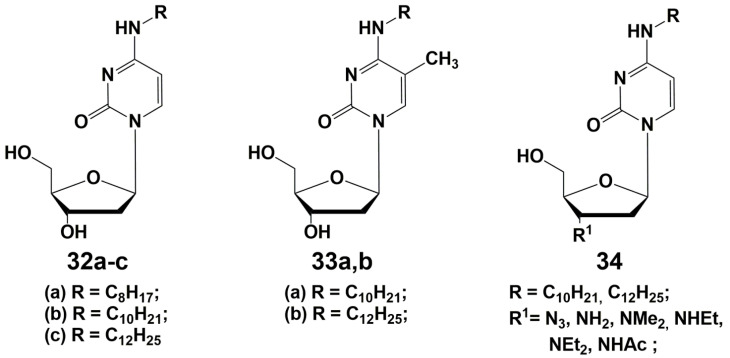
N4-Alkyl derivatives of 2′-deoxycytidine and 2′,3′-dideoxy-3′-modified cytidine.

**Figure 10 microorganisms-10-01299-f010:**
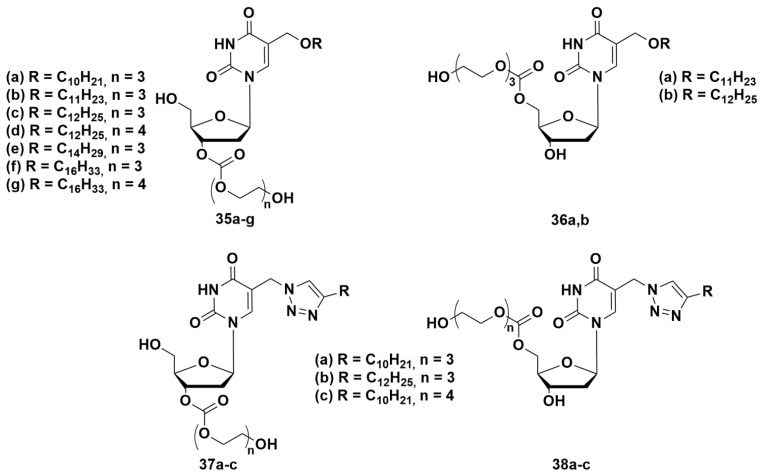
Depot forms of 5-alkyloxymethyl—**35**, **36** и 5-alkyltriazolylmethyl; **37**, **38** 2′-deoxyuridines.

**Figure 11 microorganisms-10-01299-f011:**
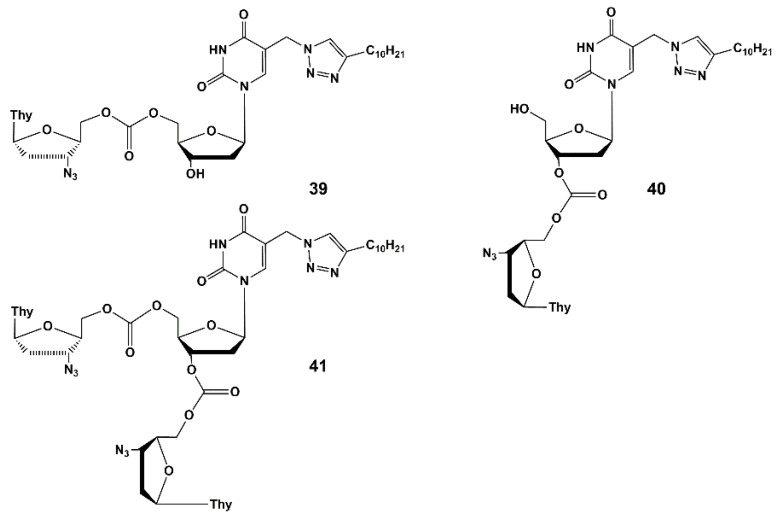
Heterodimers of AZT and 5-(4-decyl-1,2,3-triazol-1-yl-methyl)- 2′-deoxyuridine (**39**–**41**) and the 5′-norcarbocyclic analogues of 2′,3′-dideoxy-2′,3′-didehydrouridine (**42**–**43**).

**Figure 12 microorganisms-10-01299-f012:**
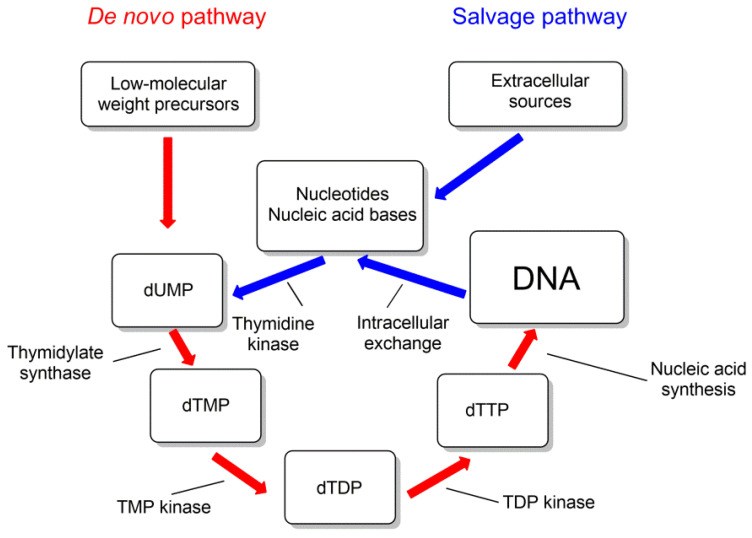
Metabolism of thymidine nucleotides.

**Figure 13 microorganisms-10-01299-f013:**
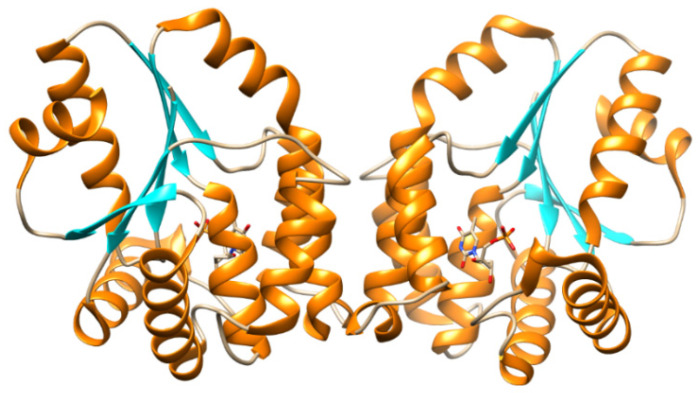
Structure of TMPK^mtb^–TMP complex (protein is presented in the form of a ribbon [82]. https://www.rcsb.org/structure/1N5I, (accessed on 17 January 2022).

**Figure 14 microorganisms-10-01299-f014:**
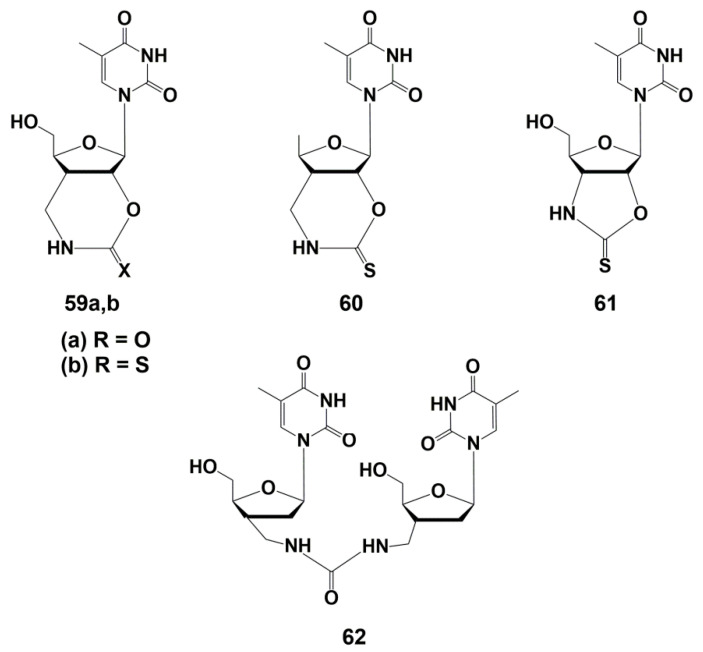
Bicyclic inhibitors of TMPK^mtb^ **59**–**62** and dinucleoside inhibitor of TMPK^mtb^ **62**.

**Figure 15 microorganisms-10-01299-f015:**
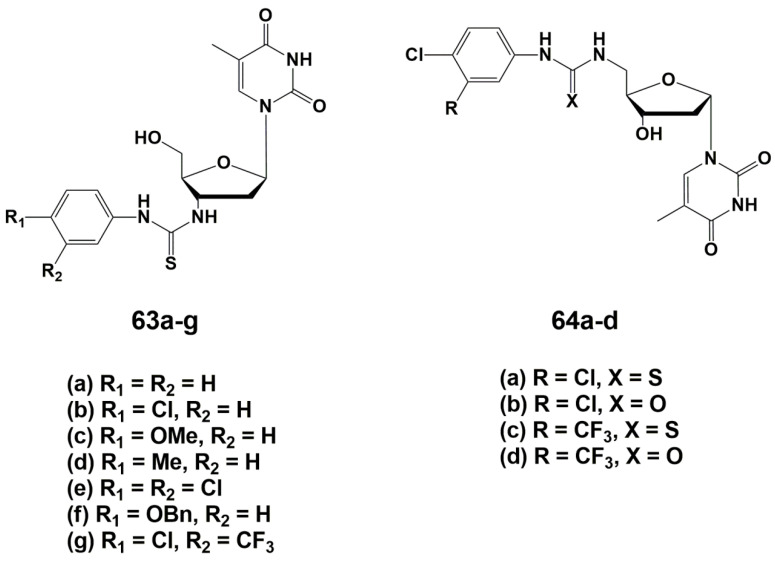
Arylurea- и arylthiourea- α- and β-thymidine derivatives **63a**–**g** and **64a**–**d**.

**Figure 16 microorganisms-10-01299-f016:**
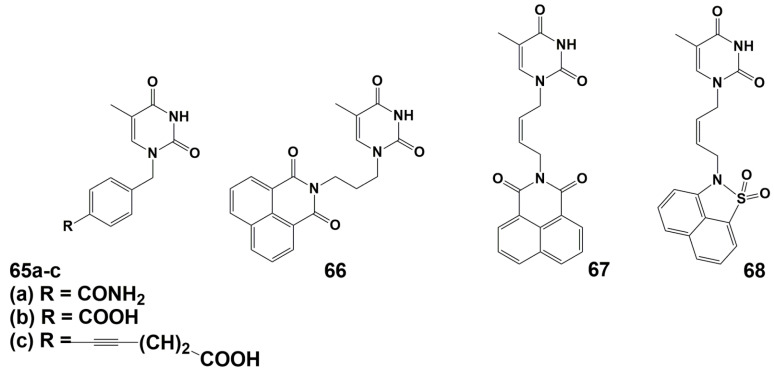
Arylpyrimidine inhibitors of TMPK^mtb^ **65**–**68**.

**Figure 17 microorganisms-10-01299-f017:**
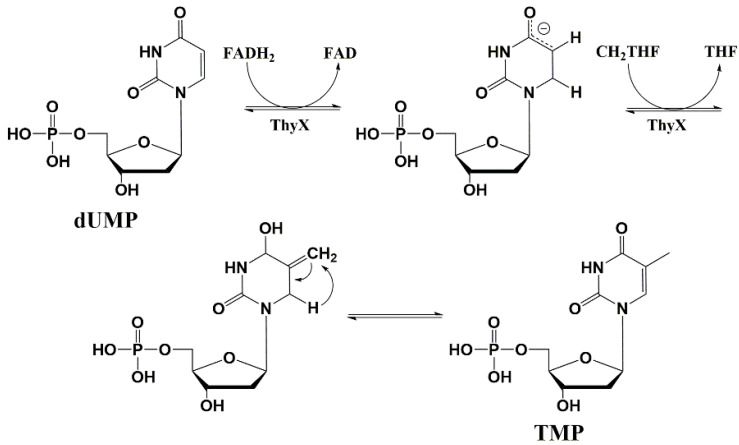
Synthesis of thymidine monophosphate from 2′-deoxyuridine monophosphate by ThyX catalysis. dUMP—deoxyuridine monophosphate, FAD—flavin adenine dinucleotide, FADH_2_—reduced form of flavin adenine dinucleotide, ThyX—flavin-dependent thymidylate synthase, CH_2_THF—5,10-methylenetetrahydrofolate, THF—tetrahydrofolate, TMP—thymidine monophosphate.

**Figure 18 microorganisms-10-01299-f018:**
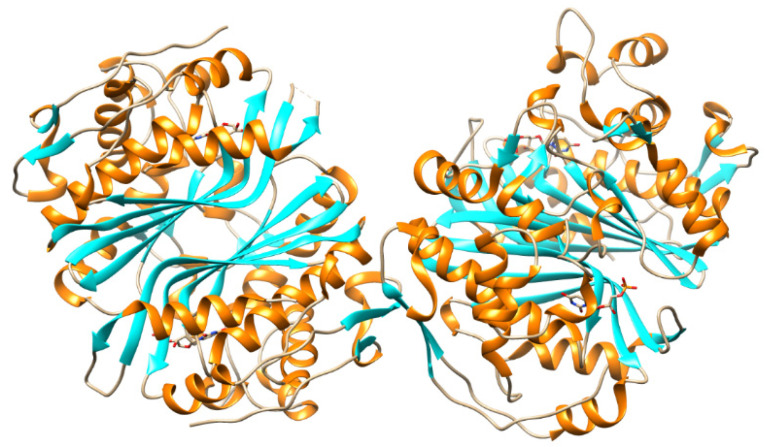
Structure of ThyX with 5-F-dUMP complex [106]. https://www.rcsb.org/structure/4FOA, (accessed on 21 January 2022).

**Figure 19 microorganisms-10-01299-f019:**
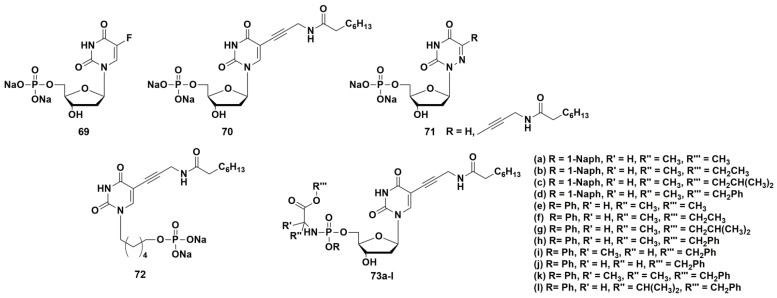
Structures of ThyX inhibitors.

**Figure 20 microorganisms-10-01299-f020:**
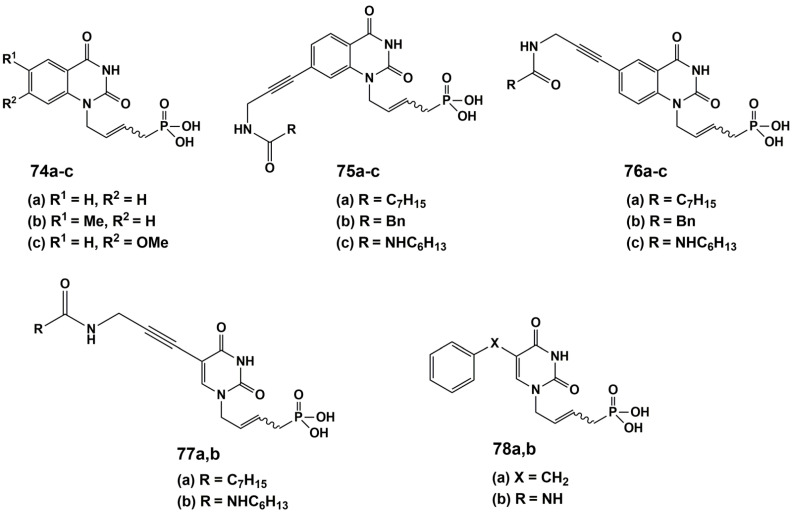
Structures of acyclic nucleoside analogues’ phosphonates—inhibitors of ThyX.

**Figure 21 microorganisms-10-01299-f021:**
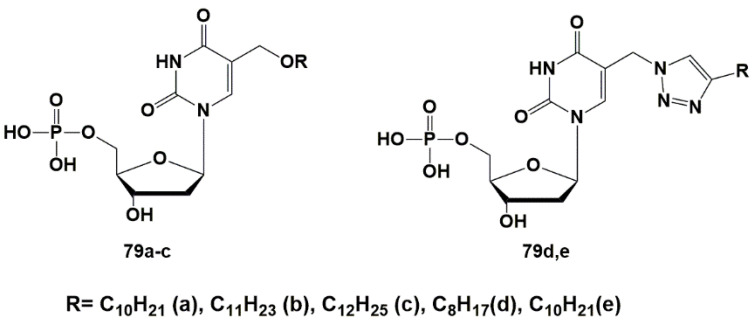
5′-Monophosphates of 5-oxymethyl and 5-triazolylmethyl-2′-deoxyuridine.

**Figure 22 microorganisms-10-01299-f022:**
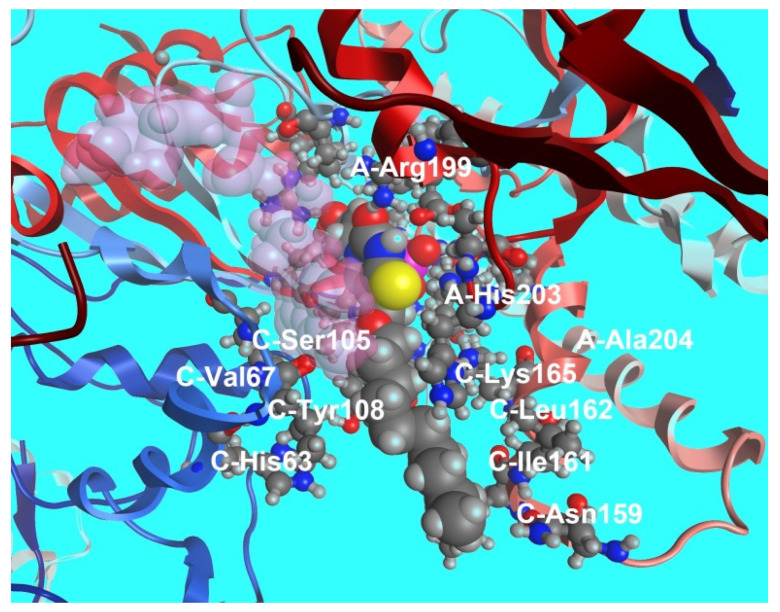
The proposed structure of the complex of 5-undecyloxymethyl-dUMP (**79b**) with ThyX (based on docking).

**Figure 23 microorganisms-10-01299-f023:**
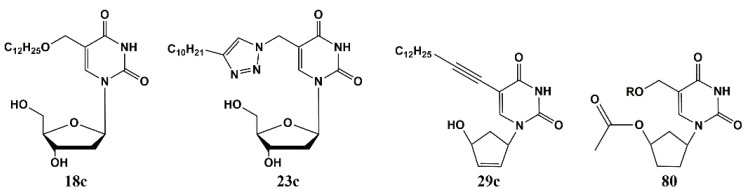
Structures of 5-substituted uracil derivatives, effective inhibitors of *Mtb* growth.

**Figure 24 microorganisms-10-01299-f024:**
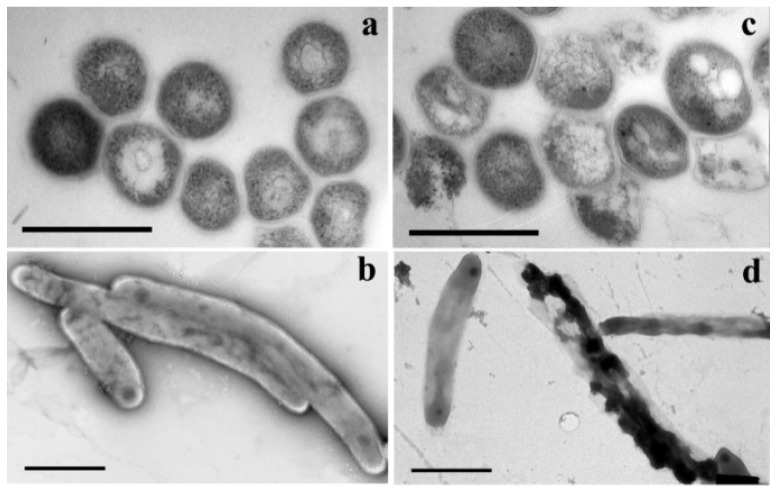
Transmission electron microscopy (TEM) of *Mtb* H37Rv cells. (**a**,**b**)—intact control cells; (**c**,**d**)—cells cultured in a medium containing the studied nucleoside derivatives; (**a**,**c**)—ultrathin sections; (**b**,**d**)—cells adsorbed on supporting film and stained with uranyl acetate. Scale (**a**,**c**)—0,5 μm; (**b**,**d**)—1 μm.

## Data Availability

Data discussed in the review are all from, and available through, cited published literatures.

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
