# Peer review of "Analogues of Pyrimidine Nucleosides as Mycobacteria Growth Inhibitors"

_microorganisms, 2022, doi:10.3390/microorganisms10071299_

Round 1

Reviewer 1 Report

First of all the authors have a huge expertise on this matter, thus making this review a high quality product.

I have just few corrections/suggestions to improve the general level of this review:

line 61: correct "may reveals"

line 69: revise "and access their ability"

line 82: change to "in the group of"

line 320 and others: check the typos (forsome)

Figure 12: correct "sourses"

table 1: the asterisk is not present in the table

line 388: change to "which proves to be"

line 391: the concentration cannot be expressed as mmol for compound 59b

Figure 15 caption: there are non English words to be revised

Figure19: for naphthyl substituent, the position must be expressed

line 473: revise "a McGuigan's group"

lines 499-500: please italicize "Mycobacterium tuberculosis"

line 502: do not italicize "ThyX"

the authors did not present in vivo studies. Are those not available for this class of compounds?

SAR studies for this scaffold must be performed for antibacterial agents and for enzyme inhibitors to help readers

line 495: this compound fairly reached the IC50 at 50 micromolar, maybe this scaffold is not so interesting for this target

Author Response

Dear reviewer #1,

We are grateful to you for reviewing our manuscript microorganisms-1772516 entitled “Analogues of pyrimidine nucleosides as mycobacteria growth inhibitors” and the comments you have made”. Here we have tried to answer them.

First of all, we would like to thank you for your appreciation of our manuscript.

We have corrected the typos you pointed out. Our corrections are marked in yellow in the text.

SAR studies for this scaffold must be performed for antibacterial agents and for enzyme inhibitors to help readers

We briefly mention SAR studies for each group of compounds in the text, and in order not to clutter up it, it seems to us inappropriate to repeat these data again. Readers, if they want a more detailed acquaintance with them, can easily refer to the literature cited by us.

The authors did not present in vivo studies. Are those not available for this class of compounds?

Unfortunately, there are practically no data in the literature on in vivo tests of any representatives of the nucleoside analogues described by us. Despite the low solubility of nucleoside analogs in aqueous solutions, in principle, their testing in vivo are quite possible. For example, mention may be made on studies of acute and chronic toxicity of N4-modified cytidine derivative 33b, which was estimated by administering intraperitoneal injections of the compound in 30% Tween-80 in white mice. The compound was found to exhibit low toxicity: the LD50 of 33b was found to be 270 mg / kg for acute administration, 215 mg / kg for subchronic administration and 200 mg / kg for the delayed effect of subchronic administration [65].

Reviewer 2 Report

The authors state that; The antimycobacterial activity of nucleoside derivatives and analogs has been revealed not so 15 long ago, and a lot of studies on their antibacterial properties have been published. Despite the fact 16 that there are no clinically use drugs based on nucleoside analogs, some progress has been made in 17
this area.

The manuscript has many technical and scientific errors. For examples, in abstract, the scientific name, Mycobacterium tuberculosis and  M. tuberculosis, both types have been used. Similarly, the first line of abstract; [Tuberculosis (TB) is the oldest human infection disease] is grammatically not suitable. The manuscript many technical and grammatical errors which should be corrected. The conclusion sections of manuscript do not provide a standard direction. Docking based studies data has been provided which need MD simulation verification. Figures resolutions and captions is not satisfactory. Although abbreviations for mycobacterium tuberculosis (M.tb) have been defined in introduction section, still line 499, Mycobacterium tuberculosis has been used.

Line 12 abstract, new multiple- and extensively drug-resistant strains of M. tuberculosis were identified, these are very old term not new. Multiple drug resistant term is not common in TB, rather, multidrug resistant (MDR) is WHO recommended.

Some references should be included: "Structural and free energy landscape of novel mutations in ribosomal protein S1 (rpsA) associated with pyrazinamide resistance". Scientific reports, 2019. 9(1): p. 7482-7494; Khan, M.T., A.U. Rehaman, M. Junaid, S.I. Malik, and Wei D.Q., "Insight into novel clinical mutants of RpsA-S324F, E325K, and G341R of Mycobacterium tuberculosis associated with pyrazinamide resistance". Computational and structural biotechnology journal, 2018. 16: p. 379-387; "Insights into the Mechanisms of the Pyrazinamide Resistance of Three Pyrazinamidase Mutants N11K, P69T, and D126N". Journal of chemical information and modeling, 2018. 59(1): p. 498-508. "Pyrazinamide resistance and mutations L19R, R140H, and E144K in Pyrazinamidase of Mycobacterium tuberculosis". Journal of cellular biochemistry, 2019. 120(5): p. 7154-7166; "Structural dynamics behind clinical mutants of PncA-Asp12Ala, Pro54Leu, and His57Pro of Mycobacterium tuberculosis associated with pyrazinamide resistance". Frontiers in Bioengineering and Biotechnology, 2019. 7: p. 404; "A computational perspective on the dynamic behaviour of recurrent drug resistance mutations in the pncA gene from Mycobacterium tuberculosis." RSC Advances 11.4 (2021): 2476-2486

Author Response

Dear reviewer #2.

We are grateful to you for reviewing our manuscript microorganisms-1772516 entitled “Analogues of pyrimidine nucleosides as mycobacteria growth inhibitors” and the comments you have made”. Here we have tried to answer them.

  1. Line 12 abstract, new multiple- and extensively drug-resistant strains of  tuberculosis were identified, these are very old term not new. Multiple drug resistant term is not common in TB, rather, multidrug resistant (MDR) is WHO recommended.

We have corrected the typos and imprecise expressions. Among them, we have replaced, as you advised, on line 12 “multiple- and extensively drug-resistant strains of M. tuberculosis” on “multidrug resistant (MDR) strains of M.tb”.

  1. Figures resolutions and captions is not satisfactory.

We have increased the resolution of figures and captions.

  1. Although abbreviations for mycobacterium tuberculosis (M.tb) have been defined in introduction section, still line 499, Mycobacterium tuberculosis has been used.

We have corrected the designation “Mycobacterium tuberculosis” and “M. tuberculosis” in the entire text to “M.tb”.

  1. Some references should be included

Our review is devoted exclusively to nucleoside analogs as anti-tuberculosis compounds and, in this regard, in our opinion, the references cited by the referee on Mtb resistance to pyrazinamide are beyond the scope of the problem under discussion. However, the articles you cited are of great interest and we have included them in the introduction as an example.

Round 2

Reviewer 2 Report

The revision is fine, acceptable in its current form